



# The CMIP6 Data Request (version 01.00.31)

Martin Juckes[1,2], Karl E. Taylor[3], Paul Durack[3], Bryan Lawrence[2,4], Matthew Mizielinski[5], Alison Pamment[1,2], Jean-Yves Peterschmitt[6], Michel Rixen[7], and Stéphane Sénésis[8]

[1]Science and Technology Facilities Council, Oxfordshire, UK
[2]National Centre of Atmospheric Science, UK
[3]PCMDI, Lawrence Livermore National Laboratory, Livermore, DA, USA
[4]Departments of Meteorology and Computer Science, University of Reading, UK
[5]Met Office Hadley Centre, Exeter, EX1 3PB, UK
[6]IPSL, Sorbonne Université/CNRS/IRD/MNHN, Paris, France
[7]World Meteorological Organization, Geneva, Switzerland
[8]Centre National de Recherches Météorologiques (CNRM), Université de Toulouse, Météo-France, CNRS, Toulouse, France

**Correspondence:** Martin Juckes (martin.juckes@stfc.ac.uk)

**Abstract.** The data request of the Coupled Model Intercomparison Project Phase 6 (CMIP6) defines all the quantities from CMIP6 simulations that should be archived. This includes both quantities of general interest needed from most of the CMIP6-endorsed Model Intercomparison Projects (MIPs) and quantities that are more specialised and only of interest to a single endorsed MIP. The complexity of the data request has increased from the early days of model intercomparisons, as has the data volume. In contrast with CMIP5, CMIP6 requires distinct sets of highly tailored variables to be saved from each of the more than 200 experiments. This places new demands on the data request information base and results in a new requirement for development of software that facilitates automated interrogation of the request and retrieval of its technical specifications. The building blocks and structure of the CMIP6 Data Request (DREQ) which has been constructed to meet these challenges is described in this paper.

## 1 Introduction

Phase 6 of the Coupled Model Intercomparison Project (CMIP6) seeks to improve understanding of climate and climate change by encouraging climate research centers to perform a series of coordinated climate model experiments that produce a standardized set of output. Twenty-three independently-led Model Intercomparison Projects (MIPs) have designed the experiments and have been endorsed for inclusion in CMIP6 (Eyring et al., 2016). An essential requirement of CMIP6 is that the thousands of diagnostics generated at each centre from hundreds of simulations should be produced and documented in a consistent manner to facilitate meaningful comparisons across models. Hence, for each experiment the MIPs have requested specific output to be archived and shared via the Earth System Grid Federation (ESGF), and the CMIP6 organisers have imposed requirements on file format and metadata.

The resulting collection of output variables (usually in a gridded form covering the globe and evolving in time) and the associated temporal and/or spatial constraints on them are referred to as the CMIP6 Data Request (DREQ). The modelling centres participating in CMIP6 are now archiving the requested model output and making it available for analysis. The DREQ





is significantly more complicated that the data requests from previous CMIP phases, complexity which arises from the size of CMIP6 and the inter-relationships of MIPs. In this paper we describe that complexity, introduce the tools which were provided to capture and communicate the DREQ, provide some headline statistics associated with the DREQ, and outline some of the

problems encountered and potential solutions for future exercises.

In section 2 we put the CMIP data request in the context of previous data requests, and outline how the complexity of CMIP6 has increased the complexity of the DREQ. In section 3 the DREQ is motivated with relation to the science goals and oragnisational structure of CMIP6, and section 4 then defines the structure of the request. Section 5 describes the range of interfaces to the request. A summary and outlook for future developments are provided in section 6.

## 30  2  The Data Request in Context

In the 1990s the data request for the Atmospheric Model Intercomparison Project (the CMIP predecessor; Gates, 1992) was presented in a single text table listing the required variables, all requested as monthly means: 17 surface or vertically integrated fields, 7 atmospheric fields on 2 or 3 pressure levels and 7 zonally averaged fields as a function of pressure and latitude. In 2012, the CMIP5 (Taylor et al., 2011) request had grown to include about 1000 variables in a wide variety of spatial and

temporal sampling options, from annual means to sub-hourly values at a limited number of geographical locations. These were still effectively provided in a list (available at pcmdi.llnl.gov/mips/cmip5/requirements.html).

The DREQ builds on the methodology established to provide those lists but has been adapted and extended to deal with new challenges both in the complexity of the underlying science and in the nature of the expanding community. The transition to CMIP6 is described in Meehl et al. (2014) and Eyring et al. (2016). A central innovation is the process for endorsed Model

Intercomparison Projects (MIPs) to join CMIP6. Each MIP has an independent science team with their own science goals and objectives, but the data requirements need to be aggregated in order to enable efficient execution of the experiments by modeling groups.

The endorsed MIPs are organized by researchers with an interest in addressing specific scientific questions with the CMIP models.[1] Each MIP has described their overall science goals in a publication (see Table B1), and specified a combination of

experimental configurations and/or data requirements. The data requirements, include lists of diagnostics needed to address the science questions and specification of the experiments they are needed from. In their initial versions, the diagnostics were often not precisely defined, so refinements were made through multiple iterations to arrive at a final well defined version for the DREQ. Many experiments and outputs were shared across MIPs, leading to cross-MIP iterations around requirements and definitions. The resulting variable definitions were subsequently aggregated into a consolidated structured document, which

constitutes the DREQ and is the focus of this paper.

The challenge of the process arises from the scale and diversity of the subject matter. The 23 participating MIPs are all international consortia, some of them organized many years ago, others formed specifically for the CMIP6 exercise. The syntax

---

[1]As part of the endorsement process, each MIP must demonstrate the backing of modeling groups who will execute the numerical experiments they specify



of the technical requirements relies largely on the NetCDF Climate and Forecast Metadata (CF) Conventions[2] and builds on long-standing CMIP practice, but there were also new aspects of the technical requirements which developed dynamically over the planning stages for CMIP6 (see Balaji et al., 2018) as part of a new CMIP6 endorsement process.

Evolving requirements complicated design and implementation of the DREQ, but arose from the interconnection between the data request, the MIPs, the committees governing CMIP6, and other elements of the infrastructure described in Balaji et al., many of which were themselves evolving in response to the complexity of CMIP6. These other activities and the linkages both supported and constrained the DREQ itself.

## 2.1 The Challenge of Scientific Complexity

The complexity of climate models continues to increase (e.g. Hayhoe et al., 2017), driven by pressing societal challenges (Rockström et al., 2016). With the expanded scope of the intercomparison, and with the steadily increasing complexity of the Earth System Models, CMIP6 posed new challenges for the data request. Here we illustrate some of that complexity by considering the cryosphere, as depicted in figure 1, and then consider how this sort of complexity plays out over the data request.

The models, and hence the variables described in the DREQ, distinguish between land ice formed on land from the consolidation of snow, and sea ice formed at sea by the freezing of sea water. They have different properties, both at the microscopic scale (land ice generally contains trapped air bubbles) and at the macroscale (sea ice is typically up to a few metres thick, land ice is often hundreds of metres thick). A few of the details shown in the figure are represented for the first time, or better represented in some CMIP6 models. These include the representation of sea water extending under floating ice shelves, more detailed representation of snow on ice (with different model representations of snow on sea ice versus snow on land ice), more detailed representation of snow and other frozen precipitation, and both the representation of melt pools on sea ice and potential ice covering of those melt pools.

In the atmosphere, snow is made up of ice crystals and it is standard usage to consider "snow" as part of the atmospheric ice content. On the land surface, however, a snow-covered surface is generally understood to be distinct from an ice-covered surface. Hence, at the surface we have parameters for heat fluxes from snow to ice and rates of conversion from snow to ice (i.e. a mass flux from snow to ice). This distinction may sound obvious, but this subtle shift in the relationship between "snow" and "ice" occurring when the snow lands on the ground or on surface ice can cause confusion in technical terms.

In the CMIP5 climate simulations the boundary between land and sea was clearly defined and fixed in time, but, in at least some models, the CMIP6 ensemble introduces more complexity. For the first time, some models have a realistic simulation of floating ice shelves. These deep layers of ice form on land, but flow to cover large areas of ocean such as the Weddell Sea. The extent of the ice shelves can also, in a small number of experiments and models, vary in time. This introduces a range of possible interpretations for the boundary between land and sea: the leading edge of the ice shelf, the grounding line underneath the ice shelf, or perhaps the line where mean-sea-level intersects the surface under the ice.

---

[2]cfconventions.org

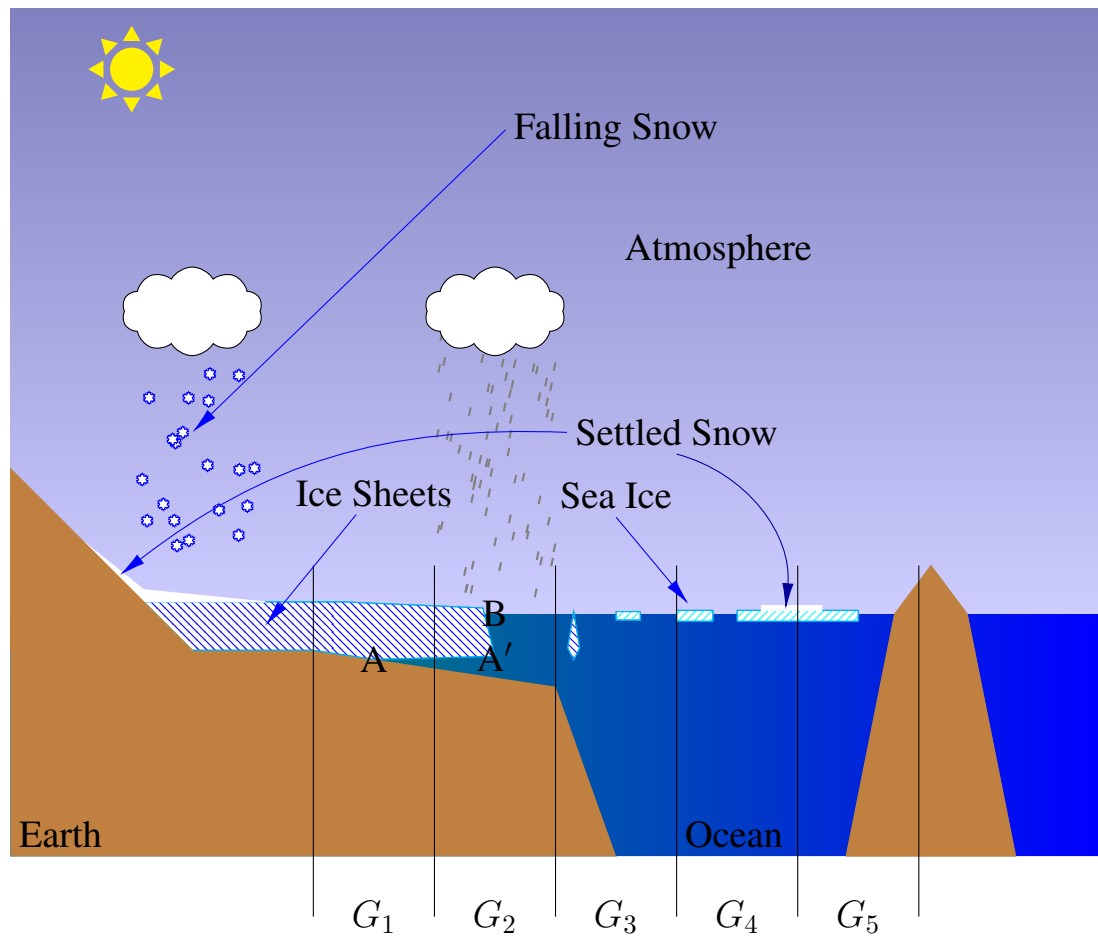

**Figure 1.** The diagram sketches a section of floating land ice and some sea ice. The vertical black lines delineate the boundaries of 5 hypothetical grid boxes. $G_1$ contains the grounding line of the ice sheet, $G_2$ contains the ice front, and $G_3$ contains some floating sea ice. $G_5$ contains a mix of ocean, sea ice and land. As models can now represent sea water extending under the ice sheet ($A$ to $A'$) there will be a difference between the grounding line ($A$) and the boundary at the ocean-atmosphere interface ($B$). In CMIP6 diagnostics, the `land` surface is taken to extend to $B$, so that diagnostics such as the surface radiation balance are treated consistently across the ice sheet surface.

In the context of CMIP6, the earth surface modelling is mainly motivated by a desire to represent energy and material cycles that affect the climate. For these purposes it generally makes sense to ignore these distinctions between grounded ice sheets, floating ice shelves, and bare land masses. Hence, for the data request, most surface `land` diagnostics are expected to extend over all land ice, including floating ice shelves. However, for a range of specialist diagnostics requested by ISMIP6, there are more specific area types defined: e.g., `grounded_ice_sheet` and `floating_ice_shelf`.

The complexities that we see in the cryosphere apply right across the domain simulated by CMIP6. Table 1 lists some of the principle categories of CMIP6 variables, showing the importance of mass fluxes and reservoirs in the overall request.





**Table 1** Categories of DREQ variables. The 2nd column shows the number of DREQ MIP variables which fall into each category. These 6 categories account for over 50% of the variables in DREQ.

| Name | Count | Comments | Units |
|---|---|---|---|
| Mass Fluxes | 274 | Fluxes of carbon (91, including 50 directly associated with carbon dioxide), water (115), nitrogen (26) and others including salt, sulphates, aerosols (42) | $\mathrm{kg\,m^{-2}\,s^{-1}}$, $\mathrm{kg\,s^{-1}}$ |
| Energy FLuxes | 153 | Radiative fluxes (83), thermal fluxes (30), parameterized heating (9), temperature tendencies (18) and various, including transports (13) | $\mathrm{W\,m^{-2}}$, $\mathrm{MJ\,m^{-1}\,s^{-1}}$, $\mathrm{K\,s^{-1}}$ |
| Mass Stores | 118 | Mass storage for carbon (49), water (43), nitrogen (17) and aerosol, sulphates etc (9) | $\mathrm{kg\,m^{-2}}$, $\mathrm{g\,m^{-2}}$, $\mathrm{kg}$ |
| Energy Stores | 30 | Stores of energy expressed as energy content or as temperature of a body | $\mathrm{J\,m^{-1}}$, $\mathrm{K}$, $^\circ\mathrm{C}$, $^\circ\mathrm{C\,kg\,m^{-2}}$ |
| Volume Fractions | 91 | A broad range of ocean tracers, most of them occurring twice: once as a variable representing the vertical structure of the volume fraction distribution and once as a single near-surface layer. | $\mathrm{mol\,m^{-3}}$ |
| Mass Fractions and Mixing Ratios | 40 | A broad range of atmospheric constituents. | "1", $\mathrm{mol\,mol^{-1}}$ |

The breakdown of variables gives a hint of the complexity that leads to such a diversity of parameters. While the headlines reports from the Intergovernmental Panel on Climate Change (IPCC) will focus on two carbon dioxide mass fluxes — from the atmosphere into the land and the ocean — but here we have 50, and a further 41 fluxes of other carbon compounds. The large

number comes from requiring representation of carbon dioxide fluxes from a range of sources (e.g. fires, natural fires, grazing, plant respiration, heterotrophic respiration[3], and crop harvesting), and masked from different land use categories (e.g. shrubs, trees, grass). Further, plant respiration is broken down into contributions from roots, stems and leaves. There are also a number of diagnostics associated with carbon isotopes $^{13}$C and $^{14}$C.

Alongside the multiplicity of variables is a multiplicity of potential applications, not all of which require the highest possible

output frequency — which is fortunate, it would be completely impractical to archive all variables at high frequency. However, this leads to the requirement of identifying, and specifying, output frequency requirements. In some cases output frequency can be reduced by carrying out processing within the simulation, so only condensed diagnostics are needed, and in others, snapshots are all that is required. In all cases, the output frequency is related to potential application objectives.

---

[3]Animals digesting plant matter





## 3 General Approach

The DREQ is designed to support a wide range of users belonging to four broad categories: the MIP science teams, modelling centers (data providers), infrastructure providers, and data users.

The MIPs contributing to CMIP6 provide input into the DREQ but also use it to coordinate their requirements with other MIPs and to obtain quantitative estimates of the data volumes associated with their planned work.

The modelling centers have two independent uses of the DREQ: first as a planning tool and second as a specification for the generation of data. When used as a planning tool, it allows exploration of the consequences of various levels of commitment in terms of data volumes and numbers of variables. When a center has begun generating data, the DREQ provides the specifications for each variable.

The main infrastructure providers who depend on the DREQ are the developers of the Climate Model Output Re-writer (CMOR) package,[4] those developing quality control software, and those doing planning for the Earth System Grid Federation (ESGF) data delivery services.[5] The relationship with the CMOR team is especially important as the DREQ and CMOR intersect in supporting the metadata specifications for CMIP6 output.

Users are mainly expected to use portal search interfaces (e.g. the ESGF search interface) to locate existing CMIP6 data, but, especially in early stages, may also rely on the DREQ to determine what data may eventually be found there.

### 3.1 Generic Requirements

The timetable for generation of the DREQ did not allow for a formal specification of technical requirements. The following list sets out the high level requirements that emerged from a range of informal discussions:

(a) Provide feedback to MIPs on feasibility of data requests, especially regarding estimated data volumes;

(b) Provide precise definitions and fully specified technical metadata for each parameter requested ;

(c) Provide a programmable interface that supports automated processing of the DREQ;

(d) Support synergies between MIPs, maximising the re-use of specifications and of data.

Item (a) is extremely important because attempting to store all variables at high frequency for all experiments would be impractical, resulting in unmanageable data volumes. Data volume estimates provided through the DREQ can only be indicative because the actual volumes will be influenced by many choices taken by modeling groups during the implementation of the request, but these estimates have nevertheless provided a useful guide for resource planning. CMIP gains immense impact from the synergies of the many science teams working on over-lapping science problems. The synergies (d) supported by the DREQ include providing standard definitions of diagnostics which can be used across multiple MIPs and making it possible for related MIPs to request output from each other's experiments.

Delivering on these led to four further technical requirements:

---

[4] cmor.llnl.gov

[5] esgf-node.llnl.gov/search/cmip6/,Williams et al. 2015





**Table 2** Choices confronting data providers within the CMIP6 Data Request

| Category | Description |
|---|---|
| MIPs supported | Data providers may opt to support one or more MIPs |
| Objectives | Some MIPs have specified different data requirements for different objectives: data providers may opt not to support all the objectives |
| Priority (variables) | Each requested variable is assigned a priority from 1 (high) to 3 (low). The priority assigned to a variable may be different for different MIPs. Data providers should supply all priority 1 variables specified for the MIPs and objectives they have specified, but may choose whether or not to supply priority 2 and 3 variables. |
| Tier (experiments) | Within each MIP the experiments proposed may be organised into tiers. Tier 1 experiments should be completed for all MIPs supported; other tiers are optional. Tiers may be assigned to experiments. There are also cases where a single ensemble member of an experiment is considered as Tier 1, and an extended ensemble as a lower Tier. There is a further complication in that MIP A may request data from an experiment defined by MIP B, but may have a different idea about the significance of that experiment to their scientific objectives. That is, an experiment defined by MIP B to be in Tier 2 may be regarded as Tier 1 or 3 by MIP A. This is dealt with by allowing the request to override the default Tier of an experiment using the `treset` attribute of a `requestItem` record. |
| Model Configuration | The data provider must, of course, choose a suitable model and model configuration to generate the data. The choice of model is relevant because some diagnostics only make sense when specific optional model components are included. |

(A) The utilisation of a flexible structured database rather than simple lists, with

(B) an informative human interface,

(C) an application programming interface to provide support for automation, and

(D) regular systematic checks to enforce consistency of technical information.

Many of these requirements were already recognized in CMIP5; the major advance in CMIP6 was the ability to tailor data needs to each individual experiment and its scientific goals, and the introduction of a programmable interface supporting automated process of the DREQ.

## 3.2 Completeness of Contribution

The intent of the DREQ is to provide all the information needed for a modelling group to archive variables of interest for subsequent analysis. In doing so, it must support the CMIP ethos of both facilitating intercomparison of an inclusive range of





models and addressing significant new areas of climate science. It must also facilitate contributions from both well established
and new participants.

In order to achieve this, CMIP6, following practice of earlier CMIP phases, allows participating institutions to be selective about the range of experiments they conduct and the diagnostics that they generate. This is facilitated by experiments defining various levels of priority for the variables requested. Hence, although the DREQ specifies all the variables requested for each experiment and ensures coherence in the data archive, it also allows some flexibility.

Table 2 shows the choices available to data providers that determine the scope of their contribution to the archive. Despite the flexibility, there is a minimum requirement: when a modelling centre commits to participating in a MIP, it is expected to provide all the priority 1 variables needed to address at least one of the scientific objectives of that MIP.

This approach ensures that CMIP has a large and representative model ensemble, but it also means that users who would like to have all models running the same collection of experiments and producing the same set of variables will not find the
consistency that they want. The data provided by some models will be more limited than for others.

To ensure some consistency across the CMIP archive, the DREQ is structured to provide a menu of choices defining blocks of variables with differing priorities and scientific objectives.

## 4 Structure

The data request contains an extensive range of specifications which define climate data products which will be held in the
CMIP6 archive[6]. The data products will, when generated in accordance with the full data format specifications[7], comply with the the data model of the CF Conventions (Hassell et al., 2017). The data request on its own does not provide the full format specifications, but does provide enough information for each variable to allow the automated production of compliant data files. That is, where there are multiple options available in the format specifcations, the data request determines which choices should be made for each variable.

In order to manage these specifications which are aggregated across the many participating endorsed MIPs, the specifcations themselves are required to fit within an information model, which we call the Data Request Information Model (DRIM) to distinguish it from the data model of the NetCDF files described by Hassell et al. (2017), on the one hand, and the Common Information Model documenting the experiments, simulations and models (Pascoe et al., 2019), on the other hand. The DRIM is expressed through an XSD schema (Juckes, 2018a) discussed further in section 4.2 below.

The nature of the process of establishing the CMIP6 Data Request has required that the DRIM itself evolve as information is gathered. In order to manage this process, the DRIM is constrained to stay within a pre-defined framework.





**Figure 2.** A core framework is explicitly coded into a python base class. The "Configuration" section is information such as the fact that the CMOR variables carry a *coords* attribute that can specify additional coordinate variables needed for a data variable. The "Content" data includes the specific names of the coordinate variables for each data variable.





. **Figure (2) continued:**

- F1: A set of core attributes are used to define additional attributes (see also B2); F2: A simple python script is used to manage framework information; F3: a style sheet is used to map XML configuration information (C5) into a schema document (C2); F4: The python base class has dependencies on the core attributes build in.

- C1: An excel workbook, defining the attributes used in each section of DREQ; C2: The schema is expressed as an XSD document; C3: A sample XML document which complies with the schema is constructed. This allows verification of the logical consistency of the schema and facilitates construction of the full DREQ XML document; C4: A python class is defined for each section, combining the base class with configuration information; C5: The excel workbook (C1) is converted to a structured XML document for robust portability;

- P1: An XML document contains the aggregated information content; P2: A python API provides a programmable interface and command line options; P3: Web pages support browsing and searching.

## 4.1 Building blocks of the DREQ

Figure 2 provides a schematic view of the structure that has emerged, exposing the three key sections: framework, configuration, and content.

The *framework* element is designed to be flexible and provide some basic software functionality to support the development and use of the DREQ. It specifies that the content will consist of a collection of sections, each of which contains some header information and a list of data records. Each data record is a list of key-value pairs, with a specific set of keys defined for each section. Each key is, in turn, defined by a record, as explained further in section 4.2 below.

The *configuration* provides the full specification of the sections in the DREQ and the attributes carried by records in each

section. For instance, records in the `var` section carry the attributes `uid`, `label`, `title`, `sn` (a link to a CF standard name), `units`, `unid` (an identifier for the units)[8], `description`, `provmip` (identifying the MIP responsible for initially defining the parameter), `prov` (a hint about the provenance) and `procComment` (processing guidance)[9].

The *content* of the DREQ describes what is actually requested by including specific information about parameters and requirements, such as the description of the `baresoilFrac` variable given in Table 3. Each of the attributes is assigned a

value that may be a free text string or a link to another DREQ record. The content can be accessed via several different methods (section 5).

---

[6]CMIP6 - Coupled Model Intercomparison Project Phase 6: pcmdi.llnl.gov/CMIP6.

[7]See WIP position papers (WGCM Infrastructure Panel, 2019) and CMOR documentation (Nadeau et al., 2018).

[8]The redundancy between "unid" and "units" has not yet been eliminated because in the absence of a fully developed suite of tools for managing linked content, such redundancy has some value. It allows easy reading of content (via the `units` value) as well as robust linking (via the `unid` attribute).

[9]This attribute is not fully implemented in the existing DREQ.





**Table 3** Example: attributes of the MIP variable record for *bare soil percentage area coverage*.

| | |
|---|---|
| label | baresoilFrac |
| title | Bare Soil Percentage Area Coverage |
| description | Percentage of entire grid cell that is covered by bare soil. |
| units | % |
| procnote | |
| prov | `CMIP5_Lmon, PMIP3_Lclim, PMIP3_Lmon, SPECS_Lmon` |
| unid | fd6ee984-3468-11e6-ba71-5404a60d96b5 |
| provmip | CMIP5 |
| sn | `area_fraction` |
| procComment | |
| uid | 9cdb8d54d49e98acadd87e2a1139225e |

## 4.2 Schema and Content Implementation

The reference document for the Data Request content is an XML document (Bray et al., 2008) conforming to an XML Schema Definition (Gao et al., 2008) (XSD) document. The schema has been developed to satisfy the requirements that have emerged

during the MIP endorsement process. The configuration-driven approach allows the Data Request Schema to be generated from a framework document, and the same framework document is used to generate Python classes for the Application Programming Interface (API).

The Request Document aims to be self descriptive: each record is defined by its attributes, and for each attribute there is a record defining its role and usage. The apparent circularity is resolved as shown in Table B2, where the `description`

attribute of the record defining `description` defines itself. The framework also constrains the set of value types used to define attributes. Some of these are generic types, such as "integer" or "string", others are more specialised such as "integerList", for a list of integers. There are 29 sections in the DREQ, the total number of attributes is 288. These are listed in a technical note[10] Full details are in the schema specification (Juckes, 2018a).

The DREQ is presented as a document of 33 sections, where each section has the following characteristics:

• The section is described by 8 attributes;

• Each section contains a list of records, each having a set of attributes;

• Each record attribute is defined by the properties listed in Table B2.

---

[10]Sections and Attributes:

https://github.com/cmip6dr/gmd2019/raw/master/slist.pdf.





### 4.2.1 Core Request Elements

The core DREQ sections are shown in Figure 3. Starting at the bottom left, a MIP Variable defines a physical quantity. Each

variable has a unique label, a title conforming to the style guide (Juckes, 2018b), a standard name from the CF conventions, and units of measure. The DREQ spans around more than 1200 different MIP variables, ranging from surface temperature to the properties of aerosols, microscopic marine species and a range of land vegetation types.

Each MIP Variable may be used by multiple CMOR Variables, which specialize the definition of a quantity by specifying its output frequency, coordinates (e.g. should it be on model levels in the atmosphere or pressure levels?), masking (e.g.,

eliminating all data over oceans), and temporal and spatial processing (e.g. averaging or summing). For instance, the near surface air temperature is a MIP variable, `tas`, used in 10 different CMOR variables that differ in frequency from sub-hourly to monthly and that cover different regions (e.g., global or Antarctica only). There are more than 2000 distinct CMOR variables in the DREQ.

Each MIP determines which variables are needed for their planned scientific work, and they are asked to assign to each

variable a priority from 1 to 3, with 1 being the most important, to each variable. The Request Variable section specifies variable priority on an experiment-by-experiment basis, leading to over 6000 distinct Request Variables.

The 3-level hierarchy of MIP variable, CMOR variable and Request Variable provides some flexibility to re-use concepts, improving consistency in the DREQ. The foundation is provided by standard names from the CF convention: 927 of these are used in the CMIP6 Data Request, and for 728 of these there is a unique associated MIP variable.

The Standard Name may be re-used up to 33 times, though smaller degrees of re-use are the norm (145 standard names used twice, 25 used three times). The standard name re-used most often is the `area_fraction`, which is used to represent the proportion of a grid cell associated with a particular category of surface type. These different categories are represented in an ancillary variable with standard name `area_type`. In most cases, when a standard name is re-used there will be additional CF metadata specifying details which distinguish between the different variables, such as the `area_type`. There are a handful

of cases, such as "`Upwelling Longwave Radiation [rlu]`" and "`Upwelling Longwave Radiation 4XCO2 Atmosphere [rlu4co2]`" for which the difference is only in descriptive metadata (in this case the `rlu4co2` variable uses an atmosphere with carbon dioxide levels enhanced by a factor 4).

There is a similar story with the relationship between MIP variables and CMOR variables: 857 MIP variables are associated with a unique CMOR variable, 283 have two and 57 have three. The MIP variable which is most heavily re-used is "`Air`

`Temperature [ta]`", with 18 associated CMOR variables. The CMOR variables are distinguished by properties such as frequency, spatial masking and temporal processing (e.g. time mean versus instantaneous values). Finally, 1120 CMOR variables have a single Request Variable, 262 have 2, and so on, up to one which has 28 different Request Variables.

The Request Variables differ from each other in terms of the MIP requesting the data and the priority which they attach to it. For instance, "`Surface Downward Northward Wind Stress [tauv]`" is requested at priority 1 by HighResMIP

and DynVarMIP and at priority 3 by DCPP. If a modelling centre is aiming to support HighResMIP or DynVarMIP, they should treat this variable aas being at the higher priority.



**Figure 3.** Main elements of the DREQ schema. The rounded, double-edged shapes represent the core request elements (section 4.2.1) which describe the central functionality of the DREQ: linking parameter definitions to objectives and specific experiments. The orange chamferred shapes are simple lists of terms (section 4.2.2), and the green rounded boxes represent imported information (section 4.2.3).



When MIPs request data, they need to provide information about the experiments that the data is required from: we do not expect all defined variables to be provided from all experiments, as that would generate substantial volumes of unnecessary output.

The process of linking the 6423 Request Variables to the hundreds of experiments is structured by first aggregating the Request Variables into 272 variable groups. Modelling centres should be able to identify the scientific objectives being supported by the data they distribute. This is done through a request Link record that associates a variable group with one or more objectives and a collection of request items.

The request items link to one or more experiments and specifies the ensemble size and, optionally, a specified temporal sub-set of the experiment for the requested output from that experiment.

### 4.2.2 Simple lists

The sections denoted by orange chamferred shapes in figure 3 are simple lists of terms, with no additional links to other sections.

The `units` section defines 67 different strings which are either units of measure or scale factors for non-dimensional quantities. The units of measure are largely based on SI units (Bureau International des Poids et Mesures (BIPM), 2014), with 47 being constructed from combinations of 10 SI units (m: metre, kg: kilogram, K: Kelvin, N: Newton, s: second, W: Watt, mol: mole, Pa; Pascal, J; Joule, sr: steradian). The remaining dimensional quantities make use of common extensions (day, year, degrees east and north, degrees Celcius). There are 224 non-dimensional variables, mainly representing volume mixing ratios of gases in the atmosphere, mass mixing ratios of trace elements in the ocean, mass fractions of soil composition, and percentage coverage of different area types.

The central role of the changes in atmospheric composition in the climate is shown in the fact that the most frequently used units of measure are mass fluxes ($\mathrm{kg\,m^{-2}\,s^{-1}}$: 248 variables), energy fluxes ($\mathrm{W\,m^{-2}}$: 133), and reservoirs ($\mathrm{kg\,m^{-2}}$: 113). Adherence to the SI units in the DREQ has caused problems for some who would like to follow the practise of modifying the units string to distinguish between mass fluxes of carbon and carbon dioxide by using $\mathrm{kgC\,m^{-2}\,s^{-1}}$ for the former (as used, for instance in IPCC, 2013). The DREQ retains the standard formulation if the units string (an important requirement of the CF Conventions), but allows the domain specific usage in the `title`, as in `Heterotrophic Respiration on Grass Tiles as Carbon Mass Flux [kgC m-2 s-1]` for the variable `rhGrass`.

The `cellMethods` section contains records defining string values for the `cell_methods` attribute defined in the CF Conventions. This attribute specifies the spatial and temporal processing applied in generating the archived fields. There are 67 records in this section. The most commonly used simply define a mean over a grid cell and over a time interval (`area: time: mean`, used in 492 CMOR variables) and a similar quantity restricted to ocean grid cells (`area: mean where sea time: mean`, 438). More complex cell methods strings may refer to masking by surface area types defined in the CF Conventions, such as `area: time: mean where crops (comment: mask=cropFrac)`, which is used to denote the average over land surface areas containing crops and is use in the specification of a CMOR variable, `prCrop`, representing





precipitation falling on crops. The comment within the cell methods string is used to provide users with information about the related variable `cropFrac`, which gives the percentage of a grid cell area covered by crops.

The `Time Slice` section specifies the portions of each experiment for which output is required. A set of variables requested at 3-hourly intervals are, for instance, only required from the `historical` experiment for the period 1960 to 2014, rather than the whole simulation from 1850.

The `Choices` section lists situations in which the modelling centres must make a choice between variables. There are cases, for example, where a modelling centre can choose to report a variable as a climatology if in their model it is prescribed to be the same from year to year rather than allowed to evolve over time.

The `Model Configuration` section lists model configuration options relevant to DREQ choices, such as whether the model has a time varying thickness of ocean grid layers or a time varying flux of geothermal energy through the ocean floor.

### 4.2.3 Imported Information

The DREQ sections labeled `Endorsed MIPs`, `CF Standard Name`, and `Experiment` (all green in figure 3) host structured, self-contained blocks of information which originate from external sources.

The `Endorsed MIPs` section lists the MIPs endorsed as participants in CMIP6. The CF Standard Name section lists terms which have defined meanings in the CF Conventions. These terms are used in the definition of MIP variables (as discussed in section 4.2.1). Each CF standard name has an associated canonical unit defining its dimensionality. The units associated with the MIP variables do not need to be identical to the canonical units of the standard name, but they do need to be consistent. For example, if the canonical unit is m (meter), then units of nm (nanometer) or km (kilometer) would be acceptable, but an angular distance in radians would not.

The `Experiment` section contains information imported from experiment descriptions formalised by ES-DOC (Pascoe et al., 2019) and from the CMIP6 Controlled Vocabularies (CVs)[11]. The CVs serve as the reference source for such things as experiment names, model names, institution names. The CVs make it possible to uniquely identify various elements within CMIP and unambiguously gain access to associated information information, such as start and end dates and ensemble sizes. Such information is required to generate data volume estimates. There are a number of experiments for which requirements vary across different priority tiers (see Table 2). For example, the `land-ssp126` experiment is requested for one ensemble member at `Tier 1` and an additional two ensemble members at `Tier 2`.

### 4.3 Links and Aggregations

The DREQ can be thought of in terms of triads (or triples) linking variables, experiments and objectives. That is, whenever a variable is requested from an experiment, it is linked to one or more objectives. There are over 350,000 potential variable-experiment-objective triads in the CMIP6 Data Request, arising from various combinations of 2068 variables, 273 experiments and 93 objectives. These three-way links may be supplemented with additional information, such as a specific sampling periods or a preferred spatial grid.

---

[11]github.com/WCRP-CMIP/CMIP6_CVs





Less than 1% of the possible combinations are used, but this is still too many to manage individually, so rather than explicitly listing all these virtual triads, the Data Request organises them in groups. This results in just 411 request links, with groups of variables needed to address one or more objectives linked to groups of experiments.

Figure 4 gives a schematic view of the linkage. Each MIP may define one or more objectives. Experiments are organised into groups, with each experiment belonging to only one group. Variables are also organised into groups, but may belong to multiple groups. When a MIP requires data from only some but not all of the experiments in a group this is dealt with by linking a group of variables directly to individual experiments.

This 3-way linkage is a significant additional complexity compared to the 2-way linkage between variables and experiments

in CMIP5. While there were different parts of the CMIP5 request originating from different groups, the option for models to be run in support of particular scientific objectives is new to CMIP6.

If one looks at just the variable-experiment links, on average around 25% of all variables are requested for any one experiment. Around 80% of all variables are requested from the historical experiment. Among the variables *not* requested are decadal ocean variables sampled at high frequency and a range of variables provided only by specialized configurations of the model

(e.g., offline land-surface and ice-sheet models).

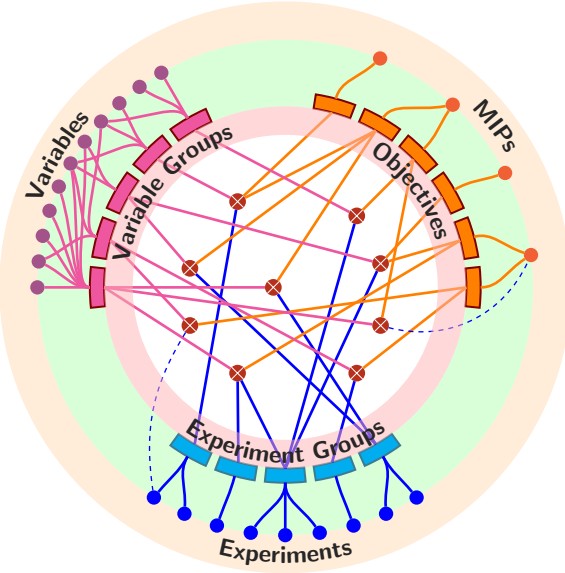

**Figure 4.** The DREQ defines a large collection of diagnostic quantities and specifies for each diagnostic, the set of experiments from which it should be provided and the objectives that it is intended to support.



### 4.3.1 Additional implicit structure

Much of the DREQ structure is formalised by use of the XSD schema mechanism, however there is a great deal more organised structure within the DREQ that is not explicitly represented by the XSD schema semantics. Prominent examples include constraints on acceptable units, the use of guide values, conditional variable requirements, and vertical domain requirements.

CF standard names have a `canonical unit`, which defines the class of acceptable units for a variable with that standard name. For instance, if the standard name has canonical unit `seconds` then associated variables can use any valid unit of time, such as `day` or `hr`. This and other consistency rules have been incorporated in the python code that, in addition to checking the schema, also checks, for example that:

- a vertical coordinate (e.g. a variable describing a property of an atmospheric layer) required by a standard name is
present;

- a cell methods string is consistent with the CF Conventions syntax rules;

- the spatial and temporal dimensions of a variable are consistent with the cell methods string (e.g. a time mean or maximum, specified in the cell methods string, requires a time dimension with a bounds attribute);

The CMIP5 request had 4 guide values for some diagnostics: minimum and maximum acceptable values and also minimum
and maximum acceptable values of the global mean of the absolute value of the diagnostic. These ranges were not intended to provide any guide to physical realism, but rather to catch data processing errors such as sign errors that might arise from institutional sign conventions opposite to those of the DREQ or incorrect units (e.g. submitting data in degrees Centigrade with metadata units describing the data as Kelvin).

With a wider range of diagnostics, for CMIP6, guide values are not always appropriate and/or available (e.g. for novel
diagnostics). The DREQ supports a three-level indication of the robustness of any specified guide values, to avoid inappropriate warnings. As an example, an analysis carried out by Ruosteenoja et al. (2017) noted that while near-surface relative humidity values of 140% can, in principle, be realistic at a point in space and time, many of the high values in the CMIP5 archive, which represented time and grid cell averages, are likely to be caused by processing errors. Hence the upper-limit is set at 100.001% and categorised as `suggested`, in contrast to the limit for sea ice extent that has a `robust` limit of 100.001%. (Excesses
over 100% are to allow for rounding errors in floating point calculations.)

The DREQ schema allows for the specification of conditionally requested variables, though this feature is not implemented for all relevant variables. For instance, there is a model configuration option `Depth Resolved Iceberg Meltwater Flux` which should be `True` for models that can represent a vertical profile of meltwater from icebergs into the ocean, and `False` if, as is the case for many models, the flux is treated as being confined to the surface. The value of this parameter then
determines whether a two or three dimensional variable should be archived to represent this flux. This feature was added in response to requests from modelling centres for a mechanism to improve automation.

Different MIPs have different requirements for data on pressure levels such as a need for zonally averaged data on 39 levels or high frequency data on 3 pressure levels. In total there are 10 different pressure axes defined as part of level harmonisation





in the DREQ (Figure 4). This harmonisation has a small cost in extra data production: for example, if one MIP is asking for a variable on 8 levels and a second MIP is asking for the same variable on 23 levels then both requests can be satisfied by providing the data on 23 levels. However, if the 23-level data is only requested for a short time period and the 8-level data is requested for the whole experiment, redundant data may be requested. This is not ideal, but it appears that the volumes of redundant data will not be excessive.

## 5 Interfaces to the Data Request

The DREQ content is provided as a version controlled XML document complying with the schema, but a range of interfaces are provided in order to make the contents more accessible. The use of XML documents ensures robust portability and allows users to import the DREQ into their own software environments.

For users who do not wish to confront the details of the XML schema, alternative views provided by the web site[12] and the python package `dreqPy`[13].

The website provides a complete view of the DREQ content in linked pages, and also a range of summary tables as spreadsheets. These include, for instance, lists of variables requested by each MIP for each experiment.

The python package provides both a command line and a programming interface. The python code is designed to be self descriptive. Every record, e.g. the specification of a variable, is represented by an instantiated class with an attribute for each property defined in the record. For example, if `cmv` is a CMOR variable record, `cmv.valid_min` will carry the value of the `valid_min` parameter for that record. The specification of the `valid_min` parameter is carried as an attribute in the parent class at `cmv.__class__.valid_min`, which is a similar instantiated class. For instance, the following code:

```
cmv.__class__.valid_min.type = "xs:float"
```

gives the data type of the attribute and `cmv.__class__.valid_min.title` gives a short description.

## 6 Summary and Outlook

The CMIP6 data request, or DREQ, provides a consolidated specification of the data requirements of the 23 endorsed MIPs[14] participating in the CMIP6 process. In doing so, it supports those responsible for configuring simulation output, those developing software infrastructure, and those who are trying to anticipate what may be available before it appears in catalogues. The latter include both those responsible for storage systems, and potential data users.

The data request has a complex structure which arises from the inherent complexity of the problem: not only are there many more MIPs and experiments than previous CMIP exercises, but not all modelling centres expect to address all the objectives of individual experiments, let alone all MIPs. This means that the request infrastructure has to handle varying aggregations

---

[12]w3id.org/cmip6dr/browse.html

[13]Data Request Python API: proj.badc.rl.ac.uk/svn/exarch/CMIP6dreq/tags/latest/dreqPy/docs/dreqPy.pdf

[14]Table B1 has 25 rows because it also includes "DECK" and "CMIP", which refer to activities that have a role analogous to MIPs in the DREQ: "DECK" specifies a collection of experiments and "CMIP" specifies a set of data requirements.





**Table 4** The pressure levels used for atmospheric variables in the DREQ.

| Pressure (hPa) | | plev39 | plev27 | plev23 | plev19 | plev8 | plev7h | plev4 | plev3h | plev3 | plev7c | single |
|---|---|---|---|---|---|---|---|---|---|---|---|---|
| 0.03 | 0.05 | ■ | | | | | | | | | | |
| 0.07 | 0.1 | ■ | | | | | | | | | | |
| 0.15 | 0.2 | ■ | | | | | | | | | | |
| 0.3 | 0.4 | ▦ | | ▦ | | | | | | | | |
| 0.5 | 0.7 | ■ | | | | | | | | | | |
| 1 | 1.5 | ■ | | ▦ | ▦ | | | | ▦ | | | |
| 2 | 3 | ▦ | | ▦ | | | | | | | | |
| 5 | 7 | ▦ | | ▦ | ▦ | | | | | | | |
| 10 | 15 | ■ | | ▦ | ▦ | | | | ▦ | | | ▦ |
| 20 | 30 | ▦ | | ▦ | ▦ | | | | | | | |
| 50 | 70 | ▦ | | ▦ | ▦ | ▦ | ▦ | | | | | |
| 80 | 90 | ■ | | | | | | | | | ▦ | |
| 100 | 115 | ■ | ▦ | ▦ | ▦ | | | | ▦ | | | ▦ |
| 125 | 130 | ■ | ■ | | | | | | | | | |
| 150 | 170 | ■ | ▦ | ▦ | ▦ | | | | | | | |
| 175 | 200 | ▦ | ■ | ▦ | ▦ | | | | | | | ▦ |
| 220 | 225 | | ■ | | | | | | | | | ■ |
| 245 | 250 | ▦ | ▦ | ▦ | ▦ | ▦ | ▦ | ▦ | | ▦ | ■ | |
| 300 | 350 | ▦ | ■ | | | | | | | | | |
| 375 | 400 | ▦ | ▦ | ▦ | | | | | | | ■ | |
| 450 | 500 | ▦ | ▦ | ▦ | ▦ | ▦ | | | | ▦ | ▦ | ▦ |
| 550 | 560 | | ■ | | | | | | | | | ■ |
| 600 | 620 | ▦ | | ▦ | ▦ | ▦ | | | | | ■ | |
| 650 | 700 | ▦ | ■ | ▦ | ▦ | ▦ | | | | | | |
| 740 | 750 | | ■ | | | | | | | | ■ | |
| 775 | 800 | | ■ | | | | | | | | | |
| 825 | 840 | | ■ | | | | | | | | | ■ |
| 850 | 875 | ▦ | ■ | ▦ | ▦ | | | ▦ | | ▦ | | |
| 900 | 925 | ▦ | ▦ | ▦ | ▦ | | ▦ | ▦ | | | ▦ | |
| 950 | 975 | | ■ | | | | | | | | | |
| 1000 | | ▦ | ▦ | ▦ | ▦ | ▦ | | | | | | ▦ |





---

**Table (4) continued:** The right hand column, headed "single" contains pressure levels used for single level variables. Other columns represent collections of levels used as a vertical axis for a range of requested parameters. Black rectangles indicate a level which is occurs in only one column. The `plevc` axis is a special case that is used specifically to match diagnostics from theInternational Satellite Cloud Climatology Project (ISCCP: WCRP, 1982) cloud simulator. The three black rectangles in the "single" column, at 220, 560 and 840 `hPa` are also ISCCP levels.

---

across the over 350,000 potential combinations of variables, experiments, and objectives, and deliver the appropriate metadata information, lists, and summaries for the groupings which arise. In practice 411 groups are needed to serve the objectives which have been extracted from the experiment definitions.

The design of the data request delivers a separation of concerns between a request framework, configuration which specifies the sections and attributes of the request, and the actual content. In each domain (framework, configuration, content) there are information components (schema, instances) and code to support the use of that information.

### 6.1    Challenges Arising

Resolving the original ambiguities and errors in the specifications of diagnostics has resulted in frequent updates to the DREQ
documents that, although cleanly version controlled, caused significant delays and inconvenience for those attempting to begin simulations as the output configuration was changing. Most of these arose not from the data request machinery, but upstream in the definitions of the MIPs, experiments, and output requirements.

     The formal schema developed for CMIP6 establishes a robust structure, but it has some clear limitations. There are a number of rules governing the content which are not captured by the schema, and arise from a semantic mismatch between the notion
of a variable, and it's implementation in the CF conventions for NetCDF. For example, certain `cell methods` strings, such as `time: mean`, require specific forms of dimensions or coordinates.

     There are also issues around variable definitions, both in the data request, and in the conventions themselves. For example, variable names containing abbreviated references to parts of the variable definitions (e.g. "sw" for "shortwave", "lw" for "longwave") lead to both inconsistency and transcription errors. Similarly, some CF Standard Names encode information about
the nature of physical quantities and the relationships between them. However, there are variations in the syntax (e.g. variables relating to nitrogen mass may contain either `nitrogen_mass_content_of_` or `_mass_content_of_nitrogen` in the standard name) which obscure some of this rich information.

### 6.2    Technical Outlook

There are a number of areas where technical improvements can be made to support future CMIP activities and, potentially,
related work outside CMIP.

     As discussed in 4.2.3 above, there are a number of areas where the DREQ intersects with ES-DOC and CVs. There is room for closer semantic alignment, as well as some streamlining of information flow between the MIP teams and those developing



the technical documents and infrastructure. Significant overlaps with ES-DOC occur in the definitions experiments, potential model configurations, conditional variables, and objectives. Some further rationalisation of the interfaces between ES-DOC, the Data Request and the controlled vocabularies prior to new experiment and MIP design will aid all parties.

More use of re-usable and extensible lists is also anticipated. One obvious way forward would be to aim for future MIPs being able to exploit existing and re-usable variable lists, either as is, or with managed extensions.

The data request is complicated, and establishing and upgrading the content of different components requires different communications approaches. This can be seen by comparing just two of the many components:

- The `grids` section defines some technical parameters used by community software tools. The priority here is to communicate clearly with the relatively small collection of software developers to ensure DREQ updates can be supported by the software to deliver the required outcomes in terms of data structures.

- The definition of parameters in the `var` section requires a discussion among a broad range of scientific experts to reach a consensus on terminology. The definitions in this section are intended to be used by multiple MIP teams, so they must be acceptable to experts in different areas.

Upgrades to these two components are in some senses orthogonal, impacting on different groups. Further partitioning of the data request to facilitate more transparent management of request upgrades would be desirable. Such partitioning may also address complexity in the data request itself, ideally allowing more agility in its specification and use.

## 6.3 Organisational Outlook

In June 2018 a first meeting of a Data Request Support Group (DRSG) was convened with the intention of broadening the engagement in the Data Request design activities. This meeting established some objectives for future work (Juckes, 2019), covering both organisational and technical issues (some of which have been discussed in section 6.2 above).

The following this and subsequent discussions we recommend:

1. There needs to be clear guidance from the CMIP panel as to the central importance to the modelling groups of early and robust resource planning. MIPs should, early in the endorsement process, provide clear information about the expected number of simulation years needed for computation and the storage volume requirements. The infrastructure teams would then be able to monitor technical compliance with these resource envelopes as the experiment documentation and request specifications are compiled[15].

2. Endorsed MIPs should be required, as part of endorsement, to identify a technical expert responsible for liaising with, and supporting the data request.

3. Clear documentation should be in place for these technical experts so that expectations are clear as to what is required.

---

[15]The difficulties of resource estimation are compounded by the fact that the modelling groups are generally not able to predict, when the process starts, the spatial resolution of the models they will be using when the computations finally get under way.





These steps would significantly reduce bottlenecks in the preparation for future CMIP exercises, and minimise the burden on both the scientific leaders of the MIPs, and the modelling groups.

Juckes (2019) also covered some procedures which have already been implemented, including the publication of each new
version of the request as a beta version to allow time for review so that changes made match the update intentions.

## 6.4   Ongoing Importance

The entire CMIP process is predicated on producing data for analysis, informing both science and policy. The central importance of a data request to those goals is obvious, but the underlying obstacles to the construction of a well defined request are often unclear. We cannot take it for granted that the goals of participating science teams will be met without detailed attention
to output requirements, particularly when, as in CMIP, so much of the value arises from the interactions between MIPs.

This detailed attention is only going to become more important in the future as the diversity of the Earth system modelling community grows and pressure for effcent use of the computing resources needed to carry out advanced simulations and store output become greater. Getting output descriptions right will be crucial to delivering and evaluating scientific benefits, and to developing the necessary infrastructure.

The growing dependency on CMIP products by a broad sector of the research community and by national and international climate assessments, services and policy-making means that CMIP activities require substantial efforts in order to provide timely and quality controlled model output and analysis.

Although CMIP has been extraordinarily successful and leverages a large investment from individual countries, there are aspects that are fragile or unsustainable due to a lack of sustained funding. The impressive CMIP leveraging is largely due to
volunteer efforts of the research community and individual scientists who contribute to the underlying essential infrastructure.

CMIP has now reached a stage where certain components and activities require sustained institutional support for it to meet the growing expectation to support climate services, policy, and decision-making. Of particular urgency is the systematic development of forcing scenarios that require institutionalized support so that quality controlled datasets and regular updates can be provided in a timely fashion. In addition, a more operational infrastructure needs to be put in place, so that core simulations
that support national and international assessments can be regularly delivered. This includes the oversight, development and maintenance of the data requests, standards, documentation, and software capabilities that make possible this collaborative international enterprise.

A specific Resolution seeking the support of the World Meteorological Organization (WMO) to CMIP was presented and approved at the 18th World Meteorological Congress, held from 3-14 June 2019. The Resolution drew WMO Members's
attention to the importance of CMIP and its critical role in supporting the global climate agenda. Members were requested to contribute institutional, technical and financial resources as necessary to ensure sustainable and robust CMIP and CORDEX (Coordinated Regional Climate Downscaling Experiment) climate change projections delivery to the IPCC.



*Code and data availability.* The current version of the DREQ is available from the project website: w3id.org/cmip6dr under the MIT License (BSD). It is provided a versioned XML document, which can be used directly or programmatically (both command line tools and a python library are provided). The exact version of the DREQ discussed in this paper (01.00.31) is available from the Zenodo repository at 10.5281/zenodo.3361640. It is also available as a package from the Python Software Foundation at pypi.org/project/dreqPy/1.0.31/.

*Author contributions.* MJ led the devlopment of the CMIP6 Data Request. KT has developed main of the underlying principles in the process of supporting CMIP5 and earlier phases of CMIP, and contributed substantially to the harmonisation and quality control of the CMIP6 Data Request. BL has contributed on the interface with ES-DOC and on the context of metadata for Earth System Models. MM and SS have provided input from the perspective of the operational climate modelling centres, and contributed significantly to the devlopement of the request by being early adopters. AP is responsible for the procedures around the CF Convention, JP and PD contributed as data coordinators for two large sections of the request, for PaleoMIP and OMIP respectively. MR has helped to establish and maintain the governance framework which facilitated the development of the request.

*Competing interests.* The authors declare that they have no conflict of interest.

Author contributions. CP represented ES - DOC in the experiment co-design, collecting information and influencing design. MJ was respon- sible for the data request. KT led the PCMDI involvement in experiment co-design. EG and BL led various aspects of ES - DOC at different times. BL and CP wrote the bulk of this paper, with contributions from the other authors. Competing interests. The authors declare that they have no conflict of interest.

*Acknowledgements.* M.N. Juckes is funded by the UK National Centre for Atmospheric Science, EU H2020 project IS-ENES3 (824084). The early stages of work were part funded by EU FP7 project IS-ENES2 (312979). K.E. Taylor and P.J. Durack are supported by the Regional and Global Modeling Analysis Program of the United States Department of Energy's Office of Science, and their work was performed under the auspices of Lawrence Livermore National Laboratory's Contract DE-AC52-07NA27344. The CMIP6 Data Request is a collaborative effort which relied on substantial effort from the MIP teams listed in Table B1. Updates and extensions to the CF Conventions and the CF standard names lists required community consensus which emerged with the help of many regular contributors to the CF discussions, especially J. Gregory. The construction of the request relied on patient input and engagement from the science teams behind the endorsed MIPs, listed in Table B1.





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



## Appendix A: CF Convention Updates

### A1 Core Convention

The CMIP6 Data Request relies heavily on the Climate and Forecast Metadata Convention (CF). A number of modifications were required either to deal with new metadata structures or to clarify the interpretation of metadata constructs employed in the past. These were all discussed on the CF discussion forum maintained by the Lawrence Livermore National Laboratory[16]. The ticket numbers given below (#152 etc) can be used to find the relevant discussions on that site.

Temporal averaging over a region specified by a time varying mask offers some particular challenges. A long discussion ("Time mean over area fractions which vary with time [#152]") established a clear protocol for expressing the concept using the `cell_methods` attribute, and clarified the usage of methods applying to multiple dimensions.

Under the CF Convention variables can refer to geographical regions either by using the name of a region from the approved list or by using an integer flag. Some wording in the conventions document was ambiguous about the validity of the latter approach: this has now been clarified to allow the use of flags ("Clarification of use of standard region names in `region` variables [#151]").

Many standard names state that additional information should be supplied in additional CF variable attributes, or impose requirements on the dimensions. Such rules are not currently checked by the CF checker, making their status in the convention ambiguous. The discussion "Requirements related to specific standard names [#153]" is still open, but has led to a proposal for a specific set of rules which have applied to the Data Request in order to ensure reasonable completeness of metadata.

A "Clarification of Conventions attribute [#76]", which was proposed long ago, has been concluded. This allows the CF convention to be used in parallel with other compatible conventions. This is required for use with the UGRID convention in CMIP6.

A long discussion on "Subconvention for associated files, proposed for use in CMIP6 [#145]" concluded by defining a subconvention which allows variables in other files to be referenced from the `cell_measures` attribute. This allows explicit referencing of grid cell areas and volumes. Such ancillary data should, according to earlier versions of the CF COnventions, but was not included in CMIP5 files because it would, for some time varying ocean grids, substantially increase data volumes.

There is an open discussion on "Extension to `external_variables` Syntax for Masks and Area Fractions [#156]" which is exploring ways of making the link between masked variables and the relevant mask clearer. With the present convention it is possible to indicate that a variable is masked by, for instance, sea ice, but there is no mechanism for identifying the specific sea ice variable used. The discussion has not reached a conclusion, so the DREQ uses an ad-hoc syntax, placing the name of the masking variable in a comment string within the cell methods string.

[16]cf-trac.llnl.gov/trac





## A2 Standard Names

665  A total of 552 new standard names were proposed for CMIP6, of which 349 were accepted. Names were rejected when existing terms, possibly in combination with area types and other metadata, can be used to meet the requirements. The new names make up 36% of the standard names used in the DREQ.

The terms span a broad range of scientific domains, with new properties of aerosols, radiation, the crysopshere (including ice shelves and dynamic floating ice sheets, sea ice, and a more detailed representation of snow packs), vegetation, atmospheric

670  dynamics, and other aspects of the climate system.

## Appendix B: Technical Tables

## B1 Experiment Collections

Table B1: Labels used for collections of experiments in the DREQ and the number of experiments and variables in each collection. $N_V$: The number of variables requested by each MIP. "Experiments defined" refers to experiments that have been designed by that MIP. "Experiments used" refers to experiments that they are requesting data from. E.g. SIMIP is a diagnostic MIP, which means that they have not defined any experiments but they are requesting data from (i.e. "using") experiments defined by others.

| Label | Title | Experiments defined (and used) | | | $N_V$ | Reference |
|-------|-------|--------|--------|--------|-------|-----------|
| | | Tier 1 | Tier 2 | Tier 3 | | |
| AerChemMIP | Aerosols and Chemistry MIP | 14(24) | 12(14) | 9(9) | 861 | Collins et al. (2017) |
| C4MIP | Coupled Climate Carbon Cycle MIP | 2(35) | 6(50) | (7) | 661 | Jones et al. (2016) |
| CDRMIP | Carbon Dioxide Removal MIP | 3(10) | 4(8) | 6(6) | 59 | Keller et al. (2018) |
| CFMIP | Cloud Feedback MIP | 6(13) | 18(20) | - | 496 | Webb et al. (2017) |
| CMIP | Coupled MIP | 7(102) | 4(128) | (55) | 830 | |
| CORDEX | Coordinated Regional Downscaling Experiment (CORDEX) | (6) | (2) | - | 32 | Gutowski Jr. et al. (2016) |
| DAMIP | Detection and Attribution MIP | 3(11) | 3(5) | 7(7) | 493 | Gillett et al. (2016) |
| DCPP | Decadal Climate Prediction Project | 9(12) | 9(9) | 5(4) | 184 | Boer et al. (2016) |
| DECK | DECK - set of standard CMIP runs | - | - | - | 0 | Eyring et al. (2016) |
| Continued on next page | | | | | | |





| Continued from previous page | | | | | | |
|---|---|---|---|---|---|---|
| DynVarMIP | Modelling the Dynamics and Variability of the Stratosphere-Troposphere System | (31) | (24) | (20) | 60 | Gerber and Manzini (2016) |
| FAFMIP | Flux-Anomaly-Forced MIP | 3(7) | 2(2) | - | 323 | Gregory et al. (2016) |
| GMMIP | Global Monsoons MIP | 1(2) | 2(2) | 3(3) | 541 | Zhou et al. (2016) |
| GeoMIP | Geoengineering MIP | 4(13) | 7(10) | - | 734 | Kravitz et al. (2015) |
| HighResMIP | High Resolution MIP | 1(6) | 4(6) | 5(5) | 900 | Haarsma et al. (2016) |
| ISMIP6 | Ice Sheet MIP for CMIP6 | 10(18) | 6(8) | 2(2) | 208 | Nowicki et al. (2016) |
| LS3MIP | Land Surface, Snow and Soil Moisture MIP | 3(10) | 16(22) | - | 616 | van den Hurk et al. (2016) |
| LUMIP | Land-Use MIP | 7(15) | 12(27) | - | 477 | Lawrence et al. (2016) |
| OMIP | Ocean MIP | 1(7) | 1(5) | 2(2) | 456 | Griffies et al. (2016); Orr et al. (2017) |
| PAMIP | Polar Amplificaton MIP | 6(7) | 7(7) | 11(11) | 185 | Smith et al. (2018) |
| PMIP | Palaeoclimate Modelling Intercomparison Project | 5(18) | (3) | - | 343 | Kageyama et al. (2018) |
| RFMIP | Radiative Forcing MIP | 8(15) | 9(11) | - | 337 | Pincus et al. (2016) |
| SIMIP | Sea Ice MIP | (25) | (16) | (7) | 98 | Notz et al. (2016) |
| ScenarioMIP | Scenario MIP | 4 | 4 | - | 0 | O'Neill et al. (2016) |
| VIACSAB | Vulnerability, Impacts, Adaptation, and Climate Services Advisory Board (VIACS AB) | (43) | (48) | (5) | 477 | Ruane et al. (2016) |
| VolMIP | Volcanic Forcings MIP | 5(9) | 2(4) | 5(5) | 295 | Zanchettin et al. (2016) |

**B2  Attribute Properties Listings**





Table B2: Listing of the properties used to define attributes in the DREQ. Each of the 20,000 records in DREQ is defined by a selection of 288 attributes, and each of these attributes is, in turn, defined through the following properties.

| Label | Title | Description | Usage |
|---|---|---|---|
| label | Record Label | A single word, with restricted character set. Specialization of SKOS prefLabel. | A short mnemonic word which is potentially meaningful but also concise and suitable for use in a programming environment. |
| uid | Record Identifier | Unique identifier | Must be unique in the DREQ. For well known concepts this may be related to the label, but for items such as simple links between concepts an a random string will be used. |
| title | Record Title | A few words describing the object. Specialization of Dublin Core title. | A short phrase, suitable for use as a section heading |
| description | Record Description | An extended description of the object/concept. Specialization of SKOS definition. | |
| useClass | Record Class | The class: value should be from a defined vocabulary. All records in the schema definition section must have class set to "___core___". | The useClass declared for an attribute can affect its interpretation in the Python package. For example, attributes labelled as "useClass=internalLink" should refer to another data request record. |
| type | Record Type | The type specifies the XSD value type constraint, e.g. xs:string. | Used in the XSD schema to constrain attribute values. |
| techNote | Technical Note | Additional technical information which can be used to specify additional properties. | |
| superclass | Superclass | States what class the property is derived from | |
| Continued on next page | | | |





| usage | Usage notes | Notes on the usage of the predicate/concept defined by this node | |
|---|---|---|---|
| | | Continued from previous page | |

## B3 Data Request Sections

Table B3: Data Request Sections: the DREQ database is split into the following sections, each taking the form of a database table with the number of records specified in column 2.

| Name | Length | Title | Comments |
|---|---|---|---|
| \_\_core\_\_ | 15 | X.1 Core Attributes | Attributes listed in Table B2 |
| \_\_main\_\_ | 288 | X.2 Main Attributes | Attributes used in the main content of the Data Request |
| \_\_sect\_\_ | 32 | X.3 Section Attributes | Attributes used to define each section. |
| mip | 30 | 1.1 Model Intercomparison Project | Summary of information held in ES-DOC |
| var | 1273 | 1.2 MIP Variable | A definition of a physical parameter |
| CMORvar | 2068 | 1.3 CMOR Variable | Specification of file metadata for a requested parameter |
| requestVar | 6365 | 1.4 Request variable (carrying riority and link to group) | Specifying a variable, a priority and the group it belongs to |
| experiment | 273 | 1.5 Experiments | Information synchronised with CVs and ES-DOC |
| objective | 92 | 1.6 Scientific objectives | Brief statement of objectives associated with each group of variables requested |
| grids | 107 | 1.7 Specification of dimensions | |
| standardname | 4267 | 1.8 CF Standard Names | Extracted from CF standard name list |
| exptgroup | 81 | 1.9 Experiment Group | |
| Continued on next page | | | |





| Continued from previous page | | | |
|---|---|---|---|
| spatialShape | 33 | 2.1 Spatial dimensions | Different combinations of horizontal and vertical grids |
| temporalShape | 5 | 2.2 Temporal dimension | Different formulations for fixed, instantaneous, time mean and climatological means |
| structure | 226 | 2.3 Dimensions and related information | The structure records combine spatial and temporal dimensions with additional dimensions and coordinates |
| miptable | 44 | 2.4 MIP tables | |
| requestVarGroup | 272 | 3.1 Request variable group: a collection of Request Variables | |
| requestItem | 3274 | 3.2 Request Item: specifying the number of years for an experiment | |
| requestLink | 411 | 3.3 Request link: linking a set of variables and a set of experiments | |
| tableSection | 16 | 3.4 CMOR Table Sections | |
| modelConfig | 23 | 3.5 Model configuration options | |
| varChoiceLinkC | 21 | 3.6 Links a variable to a choice element | |
| objectiveLink | 614 | 3.7 Link between scientific objectives and requests | |
| remarks | 4 | 3.8 Remarks about other items | |
| varChoiceLinkR | 4 | 3.9 Links a variable to a choice element | |
| varChoice | 11 | 3.10 Indicates variables for which a there is a range of potential CMOR Variables | |
| timeSlice | 28 | 3.11 Time Slices for Output Requests | |
| qcranges | 111 | Quality Control Ranges | Guide values for physically plausible data values |
| units | 90 | Units | Units of measure |
| Continued on next page | | | |





| Continued from previous page | | | |
|---|---|---|---|
| tags | 15 | 6.1 Tags | Abbreviations used in variable technotes attribute |
| varRelations | 1 | 6.2 Relationships between CMOR variables | |
| varRelLnk | 2 | 6.3 CMOR Variable Relation Links | |
| cellMethods | 60 | 7.1 Cell Methods | Specifying spatial and temporal averaging and masking |




**Table B4.** Table showing the data volumes requested, broken down in terms of the requesting MIPs (rows) and the experiments they request data from grouped according to the MIPs defining them. Units are terabytes (T), gigabytes (B) and megabytes (MB). The second from last column gives the sum of the volumes in all other columns, and the final column gives the volume of data which is uniquely requested by the MIP associated with that row. The final row represents the aggregate volume of the combined request and, since the data requests from different MIPs overlap, this is less than the sum of the individual requests. The ScenarioMIP is unusual in that they have not directly requested data: the role here is split between ScenarioMIP specifying experiments and VIACSAB requesting data to be used in the analysis of the impacts of projected climate change in different scenarios defined by ScenarioMIP.

| | CMIP | ScenarioMIP | AerChemMIP | CDRMIP | C4MIP | CFMIP | DAMIP | DCPP | FAFMIP | GeoMIP | GMMIP | HighResMIP | ISMIP6 | LS3MIP | LUMIP | OMIP | PAMIP | PMIP | RFMIP | VolMIP | TOTAL | Unique |
|---|---|---|---|---|---|---|---|---|---|---|---|---|---|---|---|---|---|---|---|---|---|---|
| CMIP | 71T | 5.5T | 3.3T | 7.3T | 3.0T | 1.8T | 2.7T | 927G | 511G | 625G | 793G | 695G | 5.2T | 3.0T | 4.1T | 1.8T | 571G | 2.2T | 2.8T | 869G | 118T | 26T |
| VIACSAB | 22T | 39T | | | 5.2T | 5.3T | 6.6T | 11T | 442G | 2.7T | 2.2T | 388G | 3.6T | 1.8T | 2.2T | | | 3.0T | | | 106T | 66T |
| AerChemMIP | 18T | 2.8T | 37T | | | | | | | | | | | | | | | | | | 58T | 42T |
| CDRMIP | 1.3T | | | 2.7T | | | | | | | | | | | | | | | | | 4.1T | 1.4T |
| C4MIP | 12T | 9.3T | | | 20T | | 6.7T | | | 577G | | | | 9.6T | 4.4T | | | 2.0T | | | 65T | 30T |
| CFMIP | 12T | | | | | 14T | | | | | | | | | | | | | | | 27T | 25T |
| DAMIP | 9.2T | 11M | | | | | 44T | | | | | | | | | | | | | | 53T | 36T |
| DCPP | 882G | 499G | | | | | | 29T | | | | | | | | | | | | 2.7T | 33T | 28T |
| FAFMIP | 13T | | | | | | | | 3.6T | | | | | | | | | | | | 17T | 14T |
| GeoMIP | 19T | 5.7T | | | | | | | | 5.7T | | | | | | | | | | | 30T | 5.9T |
| GMMIP | 47G | | | | | | | | | | 5.5T | | | | | | | | | | 5.6T | 5.0T |
| HighResMIP | 92T | | | | | | | | | | | 83T | | | | | | | | | 176T | 163T |
| ISMIP6 | 854G | 49G | | | | | | | | | | | 1.9T | | | | | 57G | | | 2.8T | 921G |
| LS3MIP | 322G | 3.0T | | | | | | | | | | | | 44T | 1.9T | | | | | | 49T | 43T |
| LUMIP | 1.2T | 5.3T | | | 5.1T | | | | | | | | | 3.2T | 10T | | | | | | 25T | 7.7T |
| OMIP | 74T | | | | | | | | | | | | | | | 40T | | | | | 115T | 63T |
| PAMIP | 882G | | | | | | | | | | | | | | | | 12T | | | | 13T | 12T |
| PMIP | 7.0T | 534G | | | 931G | | | | | | | | | | 68G | | | 8.3T | | | 16T | 9.3T |
| RFMIP | 6.8T | | | | | | | | | | | | | | | | | | 8.0T | | 14T | 12T |
| VolMIP | 7.2T | | | | | | | | | | | | | | | | | | | 6.0T | 13T | 4.8T |
| CORDEX | 7.0T | 6.9T | | | | | | | | | | | | | | | | | | | 13T | 4.8T |
| DynVarMIP | 2.5T | 4.7T | 9.1T | | | | 3.2T | 7.2T | | | | 3.6T | | | | | | | | 4.4T | 27T | 23T |
| SIMIP | 942G | 925G | | | | | | | | | | | | | | 695G | | 862G | | | 10T | 8.8T |
| UNION | 223T | 62T | 47T | 8.7T | 26T | 18T | 53T | 45T | 4.3T | 7.7T | 7.9T | 86T | 8.3T | 51T | 15T | 42T | 13T | 13T | 10T | 12T | 757T | |