# Peer review of "The CMIP6 Data Request (version 01.00.31)"

_Geoscientific Model Development, 2019_

## Short Comment (SC1) · 14 Aug 2019

Hi

1. I think it would be helpful to name the model setup that was used to generate the data volumes in the caption of Table B4 (e.g. number of atmosphere and ocean grid points and levels).

2. In chapter 6.3 I would add the need for one version controlled homepage where everyone can find all necessary information. Explanation: For example, the homepage http://clipc-services.ceda.ac.uk/dreq/tab01_3_3.html is outdated, or, in other words, only valid for DR 1.00.29 (at least this is given in the title of the index page). All the information of this page should be accessible via one version controlled homepage such as https://github.com/WCRP-CMIP/CMIP6_CVs.

3. Typos:

   - Table 1: FLuxes
   - Line 161: the the
   - Line 236: aas
   - Line 409: communications (-s?)

Kind regards,
Christopher

---

## Short Comment (SC2) · 19 Aug 2019

Thanks for the comments.

1. Yes, the nominal model information used should be included: I will add this in.

2. This is a good suggestion. There is an overhead to maintaining multiple web sites, so meeting this requirement will have to be combined by some rationalisation of the information management. I'll add a comment.

3. Typos: thanks for pointing these out.

Regards, Martin
* * *

---

## Referee Comment (RC1) · S.A. Mickelson (Referee) · 16 Sep 2019

CMIP6 is an ambitious task of coordinating several hundreds of experiments which are being ran by over 40 modeling centers world wide. The data produced by these models are compared against each other and are eventually used to make mitigation decisions. The work described in this paper was an effort to organize the data request of each of these experiments in order to ensure that the modeling centers each provide the same set of variables with the same attributes in order to ease the process of the comparisons. This was an ambitions and necessary task that helped all centers create and format the data more easily than it had been done in the past. Also, the authors' efforts in creating several different ways to access this information made the process more smooth for several institutions who took advantage of this work and used it to automate the output for their experiments.

[Figure]

Overall this paper was well written, but it seems as if part of the motivation was lost in the introduction. While the complications of the data are highlighted and are very important to discuss, a lot of what is presented in lines 134-140 and around line number 375 seem to highlight the organization need for this work and could also be added to the introduction more clearly.

It may also be helpful to the readers of this paper if the versioning and issue tracking methods were also briefly discussed. This could probably be done within only a couple of lines.

Some technical correction suggestions:

The footnote on the bottom of page 2 should have a period

Line 64: figure needs to be capitalized

Line 161: "the" used twice

Figure 2 continued caption: under "F3:" "A" should be capitalized and in the C1:... section, there should be a period at the end instead of a semicolon

Line 198: Should there be a period after note10?

Line 236: "aas" should be "as"

Line 247: figure should be capitalized

Line 281: figure should be capitalized

In the Table 4 continued section on page 20: there should be a space between the and International

Line 467 (in author contributions): main should be many?

Line 476: "respon- sible" should be fixed

In Table B2 there seem to be inconsistencies in when a period is used at the end of a

sentence

In Table B4 the description should have gigabytes listed at "G" not "B" and megabytes as "M" not "MB"

---

## Author Comment (AC1) · 25 Sep 2019

Thank you for the review.

(1) We will add a paragraph in the introduction to highlight the new requirements in CMIP6 referred to in the 2nd paragraph of the review: the need for web and programmable interfaces, the ability to tailor the request, and the number and range of participating MIPs.

(2) We can add a paragraph on version control in section 5 (and rename this section to "Interfaces and Version Control of the Data Request").

(3) We will make the technical corrections listed.

Regards, Martin Juckes

---

## Author Comment (AC2) · 25 Sep 2019

This comment lists some typographical errors which have been drawn to my attention:

(1) The name of the last author should be Sénési, not Sénésis;

(2) Livermore (institution 3 in the author list) is in "CA", not "DA";

(3) Third author is "Paul J. Durack" (initial is missing in discussion paper);

(4) Experiment count to be updated to 295;

---

## Referee Comment (RC2) · Charlotte Pascoe (Referee) · 30 Oct 2019

I enjoyed reading this paper, it is clearly written and provides an in depth description of the complex infrastructure that has been developed to support the model output requirements of CMIP6. It has been fascinating to read about the methods that have been developed to tackle the deceptively simple objective to "define all the quantities from CMIP6 simulations that should be archived". The data request infrastructure described in this paper will be of clear benefit to the climate modelling community. It allows informed decisions to be made that balance the data implications of model inter-comparison project design with the downstream capacity to manage the data. Furthermore it facilitates clear identification of the data requirements of the MIPs on each other. The production of the data request required detailed attention over a long period. I was pleased to see that the long-term sustainability of such efforts was also addressed in this paper.

[Figure]

Some technical suggestions and corrections

P2 L22: This should be "is significantly more complicated than"

P2 L27: What do you mean by "In section 3 the DREQ is motivated"?

P3 L56: Add "the" before design

P5 L92: This makes more sense if you change "While the headlines reports" to "The headline reports".

P9 Fig 2 Caption: It would help if you begin by saying what this is e.g. "The schematic structure of the DREQ."

P10 Fig 2 Caption F1: I would add "appendix" before B2 so it is clear to readers that B2 isn't an item in Figure 2.

P10 Fig 2 Caption F4: change "build in" to "built in".

P10 L173: Add "DREQ" before "structure that has emerged". You've just been talking about the DRIM so it would be good to make it really clear to readers that you're talking about the DREQ again.

P12 L214: Should this be "Each MIP determines which CMOR variables..."?

P12 L220: You say "The Standard Name may be re-used up to 33 times,". Why 33? I'm intrigued about how you arrive at this number.

P12 L227: Add "of" between "factor" and "4".

P12 L236: typo "aas".

P13 Fig 3: The style choice to use different box shapes and border styles in this figure will ensure that it can be interpreted by readers with impaired colour vision (and those working from grey-scale print outs).

P14 L240-L245: You capitalise some things that are objects in figure 3 but not others

e.g. "Request Variables" is capitalised but "variable groups" is not. Is there a reason for that?

P14 L260: typo "formulation if units"

P14 L269: typo "and is use in the specification"

P15 L292: typo "information information"

P16 Fig 4: What are the nodes in the centre circle? (I assume they are the DREQ triples you talk about at the beginning of section 4.3) It would help to interpret this figure to know what these nodes are.

P18 L358: Add "are" between "views" and "provided"

P18 L368: Add text to say that in this instance the data type is a floating point.

P20 Table 4 Caption: typo "which is occurs"

P20 Table 4 Caption: typo "theInternational"

P21 L403: Add "of" between "definitions" and "experiments"

P21 L409: Remove the "s" from "communications"

P21 Footnote 15: At first this reads as if modelling groups aren't able to predict when the process starts. It would be clearer if that phrase came earlier e.g. "compounded by the fact that, at the start of the process, the modelling groups…"

P23 L464: Add "as" between "provided" and "a".

P23 L467: typo "KT has developed main of the…" should probably be "KT has developed many of the…".

P23 L475-478: Remove this text, it refers to a different paper.

P28 L657: typo "COnventions"

P29 Table B1: This would be clearer if the third heading was "Experiments defined (Experiments used)".

P29 Table B1 Caption: Add some text to explain the significance of whether or not numbers appear in brackets in the Tier columns.

P31 Table B2 uid Usage: typo "an a"

P32 Table B3: What are the significance of the numbers in the "Title" column?

―――――――――――――――――

---

## Author Comment (AC3) · 13 Nov 2019

Thank you for the useful and encouraging review. We will implement the changes suggested when revising the manuscript.

Concerning the questions raised:

- The statement "In section 3 the DREQ is motivated .." refers to the presentation of the factors motivating the data request. We will clarify the text.

- P12, L214: yes, this does refer to CMOR variables.

- P12, L220: 33 is just the maximum number that occur within the CMIP6 implementation, it is not an imposed limit. We will rephrase to avoid the mis-leading suggestion.

[Figure]

- P14, L240-245: capitalisation needs to be standardised as suggested;

- P16, Fig 4: the central nodes are request links – we will ad explanatory text.

- P32, Table B3: the numbers in the title column are part of the title text. They were introduced in an attempt to create some organisation of the sections into chapters without modifying the underlying schema. We will add an explanatory comment.

―――――――――――――――――――――

---

## Author Comment (AC4) · 13 Nov 2019

It became clear during the final stages of manuscript submission that, given the style rules governing the presentation of table in GMD, the information in Table 4 would be better presented as a figure. In the discussion paper, the "table" has cells which are black or grey. This conflicts with the style guide (colour is not allowed in tables). One option would be to place text in the cells instead of relying on background colour, but that would require expanding the table to extend over more than one page, which would make it significantly harder for users to appreciate the information content.

Re-casting it as a figure which consists of a grid of coloured cells would avoid this. This would not involve any change to the visual presentation, it is just a matter of labelling it as a figure (comparable with Figure 2 of https://doi.org/10.5194/gmd-9-1747-2016, which uses coloured cells). When this option was appreciated it was, unfortunately, too

late to make a switch from table to figure, but we would like to make the change when revising the manuscript.

---

## Author Response (AR1)

**1 Response to SC1**

- 1. Specify model setup. This comment is accepted. "Data volumes are estimated for nominal model with 1 degree resolution and 40 levels in the atmosphere and 0.5 degrees with 60 levels in the ocean" added to caption.

- 2. Chapter 6.3. This comment is accepted. An additional recommendation added on version control of the web interface;

- 3. Four typos corrected.

**2 Response to RC1**

**2.1 General Remarks**

- A new paragraph has been added: "The challenges in the informatics domain associated with specifying a vast range of technical information are compounded by organizational and communicational challenges associated with the diverse range of stakeholders and scientific contacts, many of them in ad-hoc organizations which are themselves evolving in response to the broader CMIP challenge."

- A paragraph on version control has been added in section 5, and the section renamed to "Interfaces and Version Control of the Data Request".

**2.2 Technical Comments**

- Page 2 footnote: period added;

- Line 64: "figure" changed to "Figure";

- Line 161: duplicate "the" fixed;

- Figure 2 continued caption: capitalisation and punctuation fixed as requested;

- Line 198: period added;

- Line 236: "aas" changed to "as";

- Line 247, 281: figure capitalized;

- In the Table 4 continued section on page 20: space added between "the" and "International".

- Line 467: main changed to be many?

- Line 476: this paragraph has been deleted (see RC2).

- In Table B2: added final period for all sentences on column 2, and also in the final column of Table B3;

- In Table B4 the description: "B" replaced with "G" and "MB" with "M".

**3   Response to RC1**

- P2 L22: "that" replaced with "than";

- P2 L27: text replaced with ".. the issues motivating the DREQ are presented in the context of ..";

- P3 L56: "the" added;

- P5 L92: "While the headline reports .... will focus on ..." changed to "Although the headline reports .... generally focus on ...". The intention is not to preempt the forthcoming IPCC report, but to comment on the general practise. This change addresses the issue raised by the reviewer.

- P9 Fig 2: Caption and accompanying text clarified.

- P9 Fig 2: "see also B2" changed to "see also Table B2";

- P9 Fig 2: "build" changed to "built";

- P10 L173: Issue resolved in rephrasing of text: "The DREQ is constructed through 3 key sections, framework, configuration, and content, which are shown schematically in Figure 2."

- P12 L214: yes, "CMOR" inserted.

- P12 L220: Replaced with "The Standard Name may be re-used multiple times: 145 standard names used twice, 25 used three times. The standard name re-used most often (33 times) is ..."

- P12 L227; "of" inserted;

- P12 L236: typo fixed;

- P13 Fig 3: thank you;

- P14 L240-245: capitalisation standardised (using capitals and font);

- P14 L260: "if" replaced with "of";

- P14 L269: "use" replaced with "used";

- P15 L292: duplicate word removed;

- P16 Fig. 4: Explanation added: "The objects in the centre of the diagram represent the Request Link records which connect experiments, objectives and data specifications."

- P18 L358: "are" added.

- P18 L368: "(floating point)" inserted after "attribute".

- P20 Table 4: 2 typos corrected;

- P21 L403: "of" inserted;

- P21 L409: typo corrected;

- P21 Footnote 15: reordered as suggested;

- P23 L464: "as" added;

- P23 L467: type corrected;

- P23 L475-478: text removed;

- P28 L657: typo corrected;

- P29 Table B1: column heading reworded as suggested and additional text added in caption to clarify;

- P31 Table B2: typo corrected and text rephrased;

- P32 Table b3: Explanation added

**4 Additonal**

- PCMDI address: "DA" to "CA";

- VolMIP title in Table B1 changed to "Climatic Response to Volcanic Forcing MIP";

- "Table 4" is now presented as "Figure 5": the figure presents a grid of cells in shades of grey. This is inconsistent with the formatting rules for a table, but is allowed for figures.

- Added middle initials for co-authors Durack and Mizielinski;

- Added funding acknowledgement for Mizielinski;

- Various spelling corrections;

- Minor clarifications: inserted "from CMIP5" in "transition from CMIP5 to CMIP6" (l41-2); insert "first" before "Atmospheroc Model Intercomparison Project" (l35);

[revised manuscript text omitted]

---

## Author Response (AR2)

**1  Response to Topical Editor**

Thank you for the feedback. I have implemented the corrections, with interpretations decsribed below where they are non-trivial:

- p6, l131: I've used "infeasible", which my spell-checker prefers to "unfeasible";

- p10, l180: Changed to "parameters and requirements in attributes of metadata records";

- p17, l323: Changed to "however there is a significant amount of additional semantic structure within the DREQ that is not explicitly represented";

- p22, l463: Changed to "if the programme as a whole is to meet", to preserve the intended reference to the expectations on CMIP as a whole, rather than of individual components.

**2  Additional corrections:**

Line numbers refer to latest version:

- p3, l59: "Evolving requirements added complexity to the design ..":replaciing "complicated" with "added complexity to";

- p3, l60: Replace "DREQ, but arose from the interconnection between .." with "DREQ. These requirements arose through interactions between ..". The original longer sentence is not clear garmmatically.

- p22, l462: Change "The impressive CMIP leveraging is largely due to volunteer .." to "The impressive CMIP impact is highly dependent on volunteer ..", to avoid use of "leveraging";

- various: "center" and "meter" replaced with "metre" and "centre", and additional spelling corrections.

- formating of table slightly modified due to change from "longtable" to "supertabular" package.